# Dispensing Practices of Fixed Dose Combination Controller Therapy for Asthma in Australian Children and Adolescents

**DOI:** 10.3390/ijerph17165645

**Published:** 2020-08-05

**Authors:** Nusrat Homaira, Benjamin Daniels, Sallie Pearson, Adam Jaffe

**Affiliations:** 1Discipline of Paediatrics, School of Women’s and Children’s Health, Faculty of Medicine, The University of New South Wales, Sydney 2052, Australia; a.jaffe@unsw.edu.au; 2Respiratory Department, Sydney Children’s Hospital, Sydney 2031, Australia; 3Medicines Policy Research Unit, Centre for Big Data Research in Health, Sydney 2052, Australia; b.daniels@unsw.edu.au (B.D.); sallie.pearson@unsw.edu.au (S.P.)

**Keywords:** asthma controller, dispensing pattern, children

## Abstract

The Australian Asthma Handbook does not recommend use of fixed dose combination (FDC) controller medicines for asthma in children aged ≤5 years. FDCs are only recommended in children and adolescents (aged 6–18 years) not responding to initial inhaled corticosteroid (ICS) therapy. Using Pharmaceutical Benefits Scheme dispensing claims from 2013–2018, we examined the annual incident FDC dispensing and the incident FDC dispensing without prior ICS up to 365 days. We also determined cost of FDCs to government and patients. During 2013–2018, there were 35,635 FDC initiations and 31,368 (88%) did not have a preceding ICS dispensing. The annual incidence of FDC dispensing declined from 14.7 to 7.2/1000 children. Incidence of FDC dispensing/1000 children without a preceding ICS declined from 2.1 to 0.5 in children aged 1–2 years, 7.2 to 1.7 in 3–5 years, 14.8 to 5.1 in 6–11 years, and 18.6 to 11.9 in ≥12years. The cost of FDCs was 7.8 million Australian dollars (AUD); of which 4.4 million AUD was to government and 3.3 million AUD was to patient. Despite inappropriate dispensing of FDCs in children aged ≤5 years, incidence of FDC dispensing and more importantly incidence without a preceding ICS is declining in Australia.

## 1. Introduction

Asthma is the most common chronic childhood disease. In Australia, the prevalence of childhood asthma is higher than many other high-income countries [1,2,3]. It is estimated that 20.8% of Australian children aged 0–15 years have ever been diagnosed with asthma, while 11.3% of children have a current diagnosis [4]. The annual national hospitalisation rate for this disease is 495/100,000 children aged 0–14 years [5], costing the Australian health care system ~$200 million [6]. This high burden of asthma is in part due to variation in the clinical management of asthma resulting in low value care [7]. The appropriate management of asthma includes correct diagnosis, asthma self-management education, removal of modifiable triggers, and appropriate medication.

Several national and international guidelines for the management of paediatric asthma have been created in an attempt to reduce variability, standardise clinical care across different health care providers, and to improve health outcomes for patients. Physicians across Australia are encouraged to use the freely available Australian Asthma Handbook developed by National Asthma Council, Australia [8]. The Australian Asthma Handbook (AAH version 1 and 2) does not recommend the use of inhaled fixed dose combination (FDC) controller medicines, which include a combination of inhaled corticosteroids and a long acting β2-agonists (LABAs), in children aged ≤5 years [8]. Additionally, AAH recommends use of FDCs in children ≥6 years only as a step-up controller therapy if the initial use of daily inhaled corticosteroid (ICS, anti-inflammatory) fails to control symptoms. Prior to 2019, the international Global Initiative for Asthma (GINA) guidelines recommended increasing daily dose of ICS as a step-up controller therapy in children aged 6–11 years and use of FDCs as a step-up controller therapy after an initial trial with ICS only in adolescents (≥12 years) [9].

Data from the USA [10,11] and UK [12] suggest that, despite these established guidelines, the inappropriate use of FDC is actually increasing in children. A similar trend was also observed in the Australian Capital Territory [13] and in the 2014 Pharmaceutical Benefits Scheme (PBS) post-market review of medicines used to treat asthma in children [14]. However, there have been no national studies examining the dispensing pattern of FDCs since the 2014 post-market review and our understanding of how these medicines are dispensed in contemporary practice remains limited. Therefore, the objectives of our study were to assess the patterns of asthma FDC controller medicines dispensed to Australian children using a national, 10% sample of PBS dispensing data. As FDCs are the most expensive asthma controller medicines and are not recommended for children aged ≤5 years, we also aimed to calculate the cost of these medicines to the health system. We further investigated the sequence of dispensing of FDCs with the aim of determining whether or not their use adhered to the AAH step-up recommendations. These data are helpful to quantify the extent of appropriateness in asthma controller dispensing in children, with the goal of improving asthma management for children and reducing burden (including cost) on the health care services.

## 2. Methods

### 2.1. Study Design and Population

Australia has universal healthcare arrangements for all Australian citizens and eligible residents. The PBS is a program of the Australian Government that provides subsidised prescription drugs to all residents of Australia, as well as certain foreign visitors covered by a Reciprocal Health Care Agreement [15]. We conducted a population-based, retrospective cohort study using the 10% PBS sample dataset—a standardised dataset provided by Services Australia (servicesaustralia.gov.au). This 10% sample PBS dataset is a longitudinal nationally representative random sample of the PBS-eligible, Australian population. The PBS data has records of 23 million Australian citizens. The patient population for the dataset is selected for the sample based on their unique, randomly assigned Medicare ID. The data collection includes all dispensing records of prescription medicines for the sample [15].

Our study population consisted of all children and adolescents aged 1–18 years of age who were dispensed at least one FDC between January 2013 and December 2018.The PBS data set does not include details about how the diagnosis of a specific condition was made. However, FDCs are only prescribed to children with asthma. The names of FDCs that are available in Australia and were included in the analysis are listed in Table 1.

### 2.2. Statistical Methods

We calculated annual age-stratified (1–2 years, 3–5 years, 6–12 years, and >12 years age groups) incidence/1000 children per year of FDC dispensing. We estimated incident (new) use by identifying children with a dispensing record for an FDC within a given calendar year and without any dispensing of an FDC in the preceding 12 months. We further estimated incident use of FDC without any dispensing of ICS in the preceding 12 months. We used the Australian Bureau Statistics (ABS) midyear population estimates [16] for each age group, divided by 10 to correspond to our 10% sample, as the denominator for all incidence estimates.

The total cost of FDC, including cost to government and patients based on the dispensed price and patient co-payment, over the entire study period as well as for each calendar year was estimated.

### 2.3. Ethics Approval

The New South Wales Population and Health Services Research Ethics Committee granted ethics approval for this study (approval number 2013/11/494).

### 2.4. Patient and Public Involvement

The study involved analyses of routinely collected data and did not involve any direct patient participation or recruitment.

## 3. Results

### Cohort Characteristics

During 2013–2018, 31,149 children and adolescents aged 1–18 years were dispensed at least one FDC. There were 35,635 FDC initiations and 31,368 (88%) did not have a preceding ICS dispensing. The median annual number of FDC dispensing/patient (interquartile range (IQR)) was 1 [1,2,3]. For children with two or more FDC dispensing in a year, the median time between dispensing was 70 days (IQR 37–151 days). The most commonly dispensed FDC was fluticasone and salmeterol preparation (Table 1).

During 2013–2018, the overall incidence of FDC dispensing in children and adolescents declined from 14.7–7.2/1000 children (Figure 1). The incidence of FDC dispensing /1000 in children aged 1–2 years ranged between 2.6–0.6; 8.8–2.5 in children aged 3–5 years; 16.8–6.6 in children aged 6–12 years; and 19.5–13.1 in adolescents aged >12 years (Figure 1).

Incidence of FDC dispensing/1000 children without a preceding ICS dispensing was between 2.1–0.5 in children aged 1–2 years; 7.2–1.7 in children aged 3–5 years; 14.8–5.1 in children aged 6–12 years and 18.6–11.9 in adolescents aged >12 years (Figure 2).

The overall cost of FDC for 2013–2018 in our cohort was AUD 7.8 million; of which AUD 4.5 million was to the government and AUD 3.3 million was to the patient (Table 2).

## 4. Discussion

This nationally representative, population-based study suggests that while there is inappropriate dispensing of FDC in pre-school children there is a steady declining trend in the annual dispensing of FDCs across all age groups. The observed declining trend in FDC initiations for children contrasted findings from a previous study conducted in the Australian Capital Territory over a different timeframe. That study found a 12% increase in use of FDC between 2002 and 2005 [13]. Despite the observed declining trend across more recent years, if we extrapolate our estimates to the wider Australian population, a large number of children (>50,000) were initiated on FDC therapy without a prior trial of ICS. Although there is some evidence that maintenance and reliever therapy in children with budesonide and formoterol may be beneficial [17], at the time of the study there was insufficient evidence to recommend FDC before a trial of ICS in children ≥6 years [18]. Such practice was also not supported by national and international clinical practice guidelines [8]. However, in 2019 the GINA guidelines updated their recommendations and suggested use of daily low dose of ICS or FDC (budesonide-formoterol) as needed as the first line of controller therapy in adolescents [19]. It is expected that the national guidelines will also be updated to reflect this change and thus our study will provide baseline data in terms of evaluating the change in the dispensing pattern of FDCs following this change.

In our cohort, 3500 children aged ≤5 years across Australia were inappropriately [8] initiated on FDC annually which amounted to a cost of ~AUD 500,000 to the government and patients. Such inappropriate use represents wastage of health funds. Additionally, the high cost of these medicines is a significant barrier to compliance with asthma medications [20]. The most commonly dispensed FDC was the combination of fluticasone and salmeterol. This is likely because only the combination of fluticasone and salmeterol is listed on the PBS for use and reimbursed in children aged 4 years and over.

It is pleasing to note the trend in reduced dispensing of FDC during the period of the study. Whilst we could not look into the reasons for this, this time period coincides with a significant increase in education to the prescribing community about the appropriate use of FDCs by the Australian asthma peak bodies and the Australian Paediatric Respiratory Medical Group [21] following concerns of tachyphylaxis caused by long acting beta agonists [22].

In March 2018, the Pharmaceutical Benefits Advisory Committee (PBAC), an independent expert body of doctors, health professionals, health economists, and consumer representatives appointed by the Australian Government to recommend new medicines for listing on the PBS, considered a three-year evaluation report conducted following the 2014 post market review of asthma medicines in children. Following the meeting PBAC concluded that the proportion of use of FDC outside clinical guidelines remained “unacceptably” high and recommended that listing of all FDC for asthma should be streamlined authority [23]. When prescribing a streamlined authority item, a doctor needs to ensure that the prescribing of the medication is in line with the PBS restrictions criteria for the medication and is required to add the respective streamlined authority code on the prescription [24]. This recommendation was made to promote prescribing ICS as the first line of controller therapy [23]. The data from our study will help monitor the effectiveness of this policy change over time.

There are several limitations of our study. The 10% PBS sample includes the year of birth for all patients based on dates of birth that have been perturbed by up to six months to protect individual privacy. As such, some children in our cohort would have been less than one year and greater than 18 years of age. Our data contain records of asthma prevention medicine dispensing, but lack information on adherence to medicine. While non-compliance is associated with sub-optimal management of asthma symptoms [2] we cannot assess this aspect of asthma management. Our data also lack information regarding the type of prescriber, however studies suggest that >90% of asthma preventers are initiated by primary care providers [14]. Finally, our data did not include information on treatment indication and we were unable to investigate why prescribing was not in keeping with the national guidelines.

## 5. Conclusions

In conclusion, we have demonstrated that both FDC dispensing and initiation are decreasing in Australian children, which is a promising trend. However, children aged <12 years are often prescribed FDCs without an initial therapy with ICS which is inconsistent with National and International Guidelines. Clinical practice guidelines standardize clinical care, reduce wastage of health resources, and improve the value of healthcare. There is a need to understand factors associated with guidelines adherence in order to develop appropriate interventions to improve health care professionals’ awareness of guidelines and appropriate prescribing practices.

## Figures and Tables

**Figure 1 ijerph-17-05645-f001:**
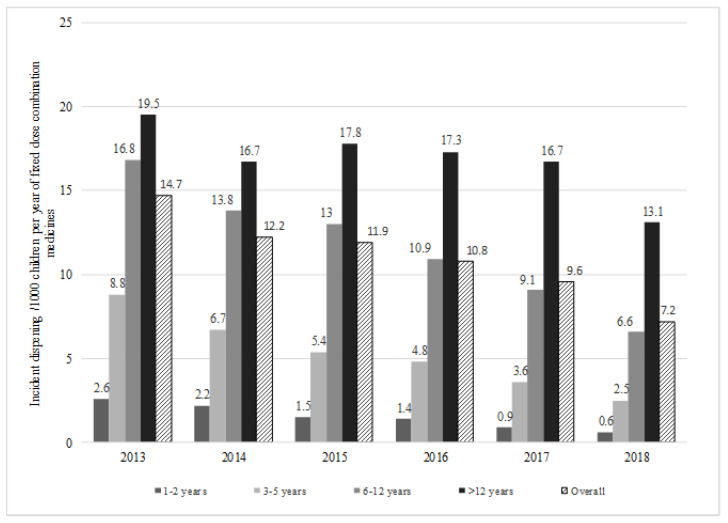
Year and age specific incident dispensing of any fixed dose combination medicines in 10% PBS sample of Australian children aged 1–18 years, 2013–2018.

**Figure 2 ijerph-17-05645-f002:**
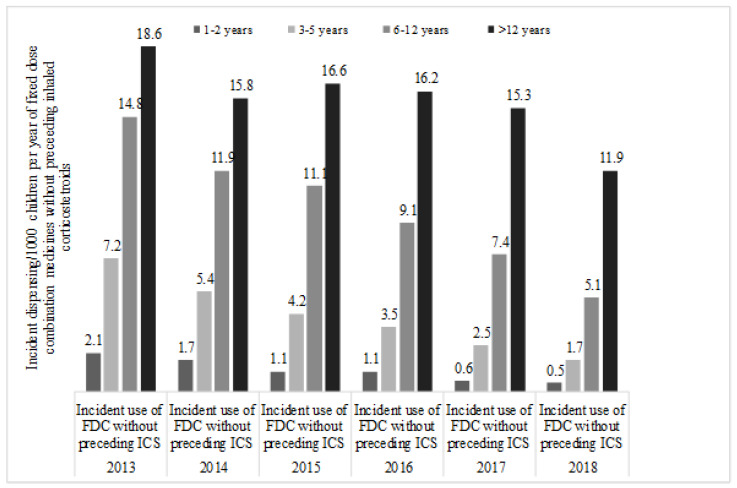
Year and age specific incident dispensing of any fixed dose combination medicine without a preceding dispensing of inhaled corticosteroid in 10% PBS sample of Australian children aged 1–18 years, 2013–2018.

**Table 1 ijerph-17-05645-t001:** Annual fixed dose combination product specific incidence/1000 children per year dispensed in 10% PBS sample of Australian children aged 1–18 years, 2013–2018.

Parameters	2013	2014	2015	2016	2017	2018
**Number of FDC Initiations**						
Fluticasone with Salmeterol	5926	4886	4534	4004	3394	2571
Budesonide with Eformoterol	1732	1517	1643	1607	1611	1173
Fluticasone with Eformoterol	0	41	95	116	110	101
Fluticasone with Vilanterol	0	2	91	117	175	189
Mid-year population of children aged 1–18 years	5,228,181	5,290,116	5,350,091	5,418,243	5,501,925	5,562,411
**Incident Dispensing of FDC Product /1000 Children Per Year**						
Fluticasone with Salmeterol	11.3	9.2	8.5	7.4	6.2	4.6
Budesonide with Eformoterol	3.3	2.9	3.1	3.0	2.9	2.1
Fluticasone with Eformoterol	0.0	0.1	0.2	0.2	0.2	0.2
Fluticasone with Vilanterol	0	0	0.2	0.2	0.2	0.3

**Table 2 ijerph-17-05645-t002:** Costs of fixed dose combination medicines by year in 10% PBS sample; 2013–2018, Australia.

Groups		Annual Cost in Australian Dollars (AUD)					
		2013	2014	2015	2016	2017	2018
Ages 1–2 years	Total costs	16,762	14,334	9048	9113	5960	2693
	Costs to government	7222	6345	4770	4255	2054	1110
Ages 3–5 years	Total costs	116,549	103,785	75,297	63,812	46,718	34,634
	Costs to government	61,263	50,521	36,650	32,323	21,479	14,535
Ages 6–12 years	Total costs	684,680	631,804	559,460	494,237	408,481	312,895
	Costs to government	398,915	365,678	306,243	266,354	207,810	155,557
Ages >12	Total costs	782,223	749,478	741,153	723,461	670,517	555,548
	Costs to government	508,807	474,827	454,674	429,425	374,020	295,064

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
