# Peer review of "Dispensing Practices of Fixed Dose Combination Controller Therapy for Asthma in Australian Children and Adolescents"

_ijerph, 2020, doi:10.3390/ijerph17165645_

Round 1

Reviewer 1 Report

Manuscript ID: ijerph-840013

Title: Dispensing Practices of Fixed Dose Combination 2 Controller Therapy for Asthma in Australian 3 Children and Adolescents

This paper examines the patterns of fixed dose combination (FDC) controller therapy among Australian asthmatic patients aged 1-18 years with large sample size. The authors used the Pharmaceutical Benefits Scheme (PBS) sample data set examining children and adolescents who were dispensed at least one FDC between 2013 and 2018. They also assessed the total cost of FDC (government and patients sides) based on the dispensed price and patient co-payment. It is an important population-based study in Australia showing inappropriate dispensing of FDC in preschool children and a steady declining trend for FDCs across four examined age groups. This is a promising study suggesting that both FDCs dispensing and initiation are declining in Australian children consistent with National and International Guidelines. The authors also reported that FDC overall cost between 3013 and 2018 was AUD 7.8 million; AUD 4.5 million to the government and 3.3 million to the patients. Such studies are important and improve asthma management and reduce the health-related costs of the governments.

I have a few minor comments to improve the manuscript as follows:

1) Methods, line 84-87:

“FDCs are the most expensive asthma controller medicines and are not recommended for children aged ≤ 5 years, we also estimated the total cost of FDC, including cost to government and patients based on the dispensed price and patient co−payment, over the entire study period as well as for each calendar year.”

I think it is a part of Aim in this study and it is better to move this sentence to the end of the last paragraph in Introduction.

2) Methods, Study design, and population: It would be better to explain more details about PBS and datasets. It is also better to include more details of asthma diagnosis in your data set such as a diagnosis by GPs or pulmonologists, conducted tests, etc.

3) Discussion, as a potential limitation in this study, it is better to mention potential selection bias in your randomly-selected dataset. Your population may not reflect the characteristics of all children in Australia.

Author Response

We are very grateful to the reviewers for their valuable comments. We have made all the necessary changes suggested by the reviewers which are in track change in the manuscript. Please find below our response to specific comments of the reviewers. We have also checked the whole document for any error in spelling and grammar.

Reviewer 1:

Methods, line 84-87:

“FDCs are the most expensive asthma controller medicines and are not recommended for children aged ≤ 5 years, we also estimated the total cost of FDC, including cost to government and patients based on the dispensed price and patient co−payment, over the entire study period as well as for each calendar year.”

I think it is a part of Aim in this study and it is better to move this sentence to the end of the last paragraph in Introduction.

Response: Thanks for your input, we have now moved this under the background section (line 59-61): “As FDCs are the most expensive asthma controller medicines and are not recommended for children aged ≤ 5 years, we also aimed to calculate the cost of these medicines to the health system”

2) Methods, Study design, and population: It would be better to explain more details about PBS and datasets. It is also better to include more details of asthma diagnosis in your data set such as a diagnosis by GPs or pulmonologists, conducted tests, etc.

Response: We have now included more information (line 75-76) “The 10% PBS dataset contains dispensing records for approximately 3.3 million Australians between 2005 and 2018.” (line80-83) “The PBS dataset does not include any diagnoses information, however, FDC asthma controller medicines are only indicated for the treatment of asthma and PBS restrictions are such that these medicines may only be dispensed to people with a diagnosis of asthma

Regarding the reviewer’s comment about prescriber information, the data set did not have information about who prescribed the medication. We had added this information in the limitation section of the original submission (Line177) “Our data also lack information regarding the type of prescriber, however, studies suggest that >90% of asthma preventers are initiated by primary care providers”.

3) Discussion, as a potential limitation in this study, it is better to mention potential selection bias in your randomly-selected dataset. Your population may not reflect the characteristics of all children in Australia.

Response: The PBS has data on all Australian citizens and our analysis was based on a 10% random sample of the PBS-eligible Australian population, hence we do not think there was a selection bias.

Reviewer 2 Report

This is an important study using a nationally representative sample. Only minor comment, it is not clear whether the author using linked data or a subsample from the PBS dataset? 

Are those children have a family history of asthma or they developed asthma after a certain age? 

Author Response

We are very grateful to the reviewers for their valuable comments. We have made all the necessary changes suggested by the reviewers which are in track change in the manuscript. Please find below our response to specific comments of the reviewers. We have also checked the whole document for any error in spelling and grammar.

Reviewer 2:

Only minor comment, it is not clear whether the author using linked data or a subsample from the PBS dataset? 

Are those children have a family history of asthma or they developed asthma after a certain age? 

Response: We thank the reviewer for the comment. Our analysis was based on a subset of the PBS data only and not linked to any other data set. PBS data set do not have information on socio-demographic characteristics, and we were not able to investigate whether children had a family history of asthma or they developed asthma after a certain age. However, the objective of the study was not to determine the risk factors (which have been investigated several times and are well accepted) but rather to determine the dispensing pattern for asthma FDC medication, for which there are no population-level data from Australia.

Reviewer 3 Report

With interest, I read the manuscript ijerph-840013. It is a very nice and elegantly written manuscript based on the solid study.

The topic addressed by the Authors is very important. The costs of health care are constantly increasing, which will continue as novel diagnostic and therapeutic methods are more and more expensive. International and national EBM-based guidelines aim to provide the patients with the most effective therapies accompanied by minimal risk of unwanted effects of the treatment. In most of the cases following guidelines is (additionally) associated with improved cost-effectiveness of the treatment. Thus, adherence to international and national guidelines is crucial.

Thus, the results presented in the manuscript ijerph-840013 are very important and should be made available to the medical community.

One facultative suggestion is that the Authors could mention that novel asthma therapies, such as antisense drugs, biologicals and others (PMID: PMID: 27155029), will definitely be associated with increased therapeutic costs although they might be able to offer more targeted and individualized approach. Therefore, optimization of the costs of the basic asthma treatment is even more important.

Author Response

We are very grateful to the reviewers for their valuable comments. We have made all the necessary changes suggested by the reviewers which are in track change in the manuscript. Please find below our response to specific comments of the reviewers. We have also checked the whole document for any error in spelling and grammar.

Reviewer 3:

One facultative suggestion is that the Authors could mention that novel asthma therapies, such as antisense drugs, biologicals and others (PMID: PMID: 27155029), will definitely be associated with increased therapeutic costs although they might be able to offer more targeted and individualized approach. Therefore, optimization of the costs of the basic asthma treatment is even more important.

Response: Thanks to the reviewer, this is indeed an interesting topic but was beyond the scope of our study. We have, however, added the following lines to the manuscript (Line 39-41): “Given the move towards treatable traits in asthma and the high costs of current biologic treatments and potential novel therapies, it is critical that the appropriate prescribing of standardless expensive asthma medications is optimized.”